Physiological responses of wild birds to artificial grass during introduction to laboratory housing

Pedro Bradley P. bradley.pedro@tufts.edu
Romero L. Michael
Department of Biology, Tufts University , Medford , MA , United States of America
Bentley George
Electronic publication date: 2025 Mar 24
Publication date: 2025
Volume: 13
Electronic Location ID: e19095
Received 2024 Aug 27; Accepted 2025 Feb 11
Copyright: ©2025 Pedro and Romero
Copyright year: 2025
Copyright holder: Pedro and Romero
License: This is an open access article distributed under the terms of the Creative Commons Attribution License, which permits unrestricted use, distribution, reproduction and adaptation in any medium and for any purpose provided that it is properly attributed. For attribution, the original author(s), title, publication source (PeerJ) and either DOI or URL of the article must be cited.
License URL: https://creativecommons.org/licenses/by/4.0/

Keywords: Stress, Corticosterone, Enrichment, House sparrow, DNA damage, Foraging

Funding: National Science Foundation (USA) IOS-1655269 This work was supported by grant IOS-1655269 from the National Science Foundation (USA) to L. Michael Romero. The funders had no role in study design, data collection and analysis, decision to publish, or preparation of the manuscript.

==============================
Introduction of wild animals to captivity induces chronic stress often leading to weight loss, increases in baseline corticosterone, and increased DNA damage. To mitigate these effects, providing enrichment to the captive environment has been proposed. Yet, studies investigating the physiological effects of captive environment enrichments are rare in wild birds. Here, we test the potential of a single enrichment factor by monitoring weight, baseline corticosterone, and DNA damage in two groups of house sparrows (Passer domesticus) during introduction to captivity: (1) birds in standard laboratory cages with food dishes and (2) birds in cages where food is spread across artificial grass to simulate a more natural foraging environment. After 3 weeks, all birds switched environments for 3 additional weeks. Weight was monitored bi-weekly while baseline corticosterone and DNA damage were measured weekly. Initially, both groups lost significant weight and weight plateaued by about 2 weeks of captivity. However, after switching housing environments, only initially grass-caged birds continued to lose weight. After one week of captivity, grass-caged birds had lower DNA damage compared to standard-caged birds. Over time, standard-caged birds remained unchanged and initially grass-caged birds increased damage after switching housing environments. There were no significant differences in baseline corticosterone across groups or over time. Our findings provide limited support for artificial grass as a substantial enrichment in mitigating the physiological consequences associated with introduction to captivity. Furthermore, given the challenges to husbandry of using artificial grass, the data are insufficiently strong to recommend the use of artificial grass as a stress-reducing enrichment to laboratory housing.

Introduction

The study of wild animals in captivity has contributed greatly to scientific discovery, especially in strengthening understanding of various aspects of animal physiology. In particular, myriad studies of wild animals in captivity have contributed to our understanding of the stress response (Romero & Wingfield, 2016). However, introduction to captivity is itself a chronic stressor (Calisi & Bentley, 2009; Fischer, Wright-Lichter & Romero, 2018). Animals housed in artificial, captive environments are exposed to potential sources of stress that they may not encounter in the wild, such as restricted housing space, changes in social groups, and increased proximity to humans (Morgan & Tromborg, 2007), and highly stressed research animals puts experimental results at the risk of being compromised. In birds, introduction to captivity induces changes in behavior (Adams et al., 2011) and decreased reproductive success (Dickens & Bentley, 2014). Physiological changes indicate chronic stress, including changes in heart rate, weight loss, increased baseline glucocorticoid release, and increases in DNA damage (Fischer, Wright-Lichter & Romero, 2018; Gormally et al., 2019a), albeit these changes differ across species and reproductive states (reviewed in Fischer & Romero, 2019).

To mitigate the physiological effects of introducing wild animals into captivity, providing enrichment to the captive environment is of great interest. ‘Enrichment’ commonly refers to improvements to the captive environment and often include more complex housing environments or housing that attempts to simulate a more natural environment (Newberry, 1995). The term ‘enrichment’ has received some critique, and alternative terms such as ‘housing supplementation’ (Benefiel, Dong & Greenough, 2005), ‘environmental refinement’ (Baumans & Van Loo, 2013), and ‘well-resourced environments’ (Cait, Winder & Mason, 2024) have been proposed. However, guidelines such as the ‘Guidelines for the use of wild birds in research’ (Fair, Paul & Jones, 2010) and regulatory bodies such as The Institutional Animal Care and Use Committee (IACUC) use the term ‘enrichment’ in their recommendations. Thus, the term ‘enrichment’ will hereafter refer to any improvements to housing that increase animal wellbeing.

Environmental enrichment has been studied extensively in mice, showing that enrichments such as adding bedding to a cage, reduces aggressive behavior (Marashi, Barnekow & Sachser, 2004), and alleviates consequences of stress (Fox, Merali & Harrison, 2006). In birds, studies have shown that ‘enriched’ housing environments alter behavior. For instance in zebra finch, adding multiple perches to cages increased locomotor activity and vocalizations, natural behaviors for these animals (Jacobs et al., 1996), increasing cage sizes reduced stereotypic behavior in European starlings (Asher et al., 2009), and adding objects to cages reduced stereotypies in canaries (Keiper, 1970). In alignment with these studies, regulatory guidelines for enrichment include providing the birds with multiple perches, ample cage sizes, ad libitum food and water and additional environmental enrichments such as adding objects to cages (Fair, Paul & Jones, 2010; Bateson & Feenders, 2010). However, these guidelines are limited by a lack of studies focusing on specific enrichment factors. Guidelines also tend to suggest additional stimuli as ‘enrichment’ although enrichments that emulate a more natural environment and allow animals to exercise natural behaviors should be considered enrichment as well (Newberry, 1995). Further, studies often focus on the effects of enrichment on behavior, including reduced stereotypic behavior (Coulon et al., 2014) or increased song production (Yamada & Soma, 2016). These studies provide valuable insight into the wellbeing of an individual, but there is also a need to understand how captivity may be altering the physiology of an individual. For instance, animals living in enriched housing environments exhibited lower baseline corticosterone compared to those not in enriched housing for both mice (Mesa-Gresa, Ramos-Campos & Redolat, 2016) and rats (Belz et al., 2003). While there is evidence in rodents that enrichments do not impair experimental outcomes or the comparability of physiological data (André et al., 2018), there is not a strong dataset for the physiological effects of enrichments in birds. There is evidence that when birds have time to habituate to a new housing environment an integrated measure of corticosterone is lower (Fairhurst et al., 2011), and birds with lower levels of baseline corticosterone had better outcomes coping with introduction to captivity (Lattin et al., 2019). Yet, there is still a need for additional studies that investigate the physiological effects of potential laboratory enrichments, so that physiologists studying birds can have a better understanding of if or how a housing environment may be impacting experimental outcomes.

In the present study, we proposed the use of artificial grass as a potential housing enrichment for wild caught house sparrows (Passer domesticus) during introduction to captivity. Cages were lined with artificial grass turf and seed was spread daily across the turf instead of being placed in a traditional food dish. In the wild, house sparrows largely forage on the ground in flocks (Lowther & Cink, 2020). The artificial grass thus increased the complexity of foraging, which is more akin to a foraging strategy these birds would use in the wild. Furthermore, studies in captive parrots have shown that manipulating the complexity of foraging strategies increased the overall activity of the bird and reduced feather picking behavior, which is thought to be a behavioral stress response (Meehan, Millam & Mench, 2003; Lumeij & Hommers, 2008). Providing grass as a foraging intervention in cockatoos increased the time spent foraging and reduced a stereotypic behavior (Fangmeier et al., 2020). Given this, we hypothesized that altering the foraging strategies of the birds in captivity by providing food on artificial grass would serve as a form of enrichment, ameliorating physiological consequences of introduction to captivity. We monitored established physiological metrics, including weight and baseline corticosterone in order to assess how well the animals adjusted to captivity. We also measured corticosterone negative feedback efficacy and baseline DNA damage in the blood. Negative feedback efficacy is an integrative measure of how well an animal can mount and shut-down a stress response, which predicts survival (Romero & Wikelski, 2010) and recovery from acute stressors (Taff, Zimmer & Vitousek, 2018). Measures of DNA damage represent a downstream consequence of elevated glucocorticoids (Flint et al., 2007). A prior study in European starlings showed that taking away enrichment can lead to worse behavioral outcomes (Bateson & Matheson, 2007), indicating that enrichments may not be suitable for temporary or transitional periods, but rather for the duration of captivity. Therefore, we additionally tested if artificial grass could serve as a temporary enrichment for the purpose of ameliorating the highly-stressful transition into captivity or if providing it temporarily would lead to worsened conditions when taken away by switching birds into standard cages after they acclimated to grass-lined cages. We predicted that captivity would increase baseline corticosterone, lead to weight loss, increase baseline DNA damage, and reduce negative feedback efficacy. We then predicted that these changes would be less extreme, or plateau sooner, in birds in the artificial grass treatment group when compared to controls. Moreover, if the grass treatment could successfully serve as a temporary, transitional enrichment, we predicted that when removed, birds would exhibit no significant changes in these physiological metrics.

Materials & Methods

Experimental design

Twenty wild adult house sparrows (15M:5F) were caught in Medford, Massachusetts, USA using mist nets in May 2022. Upon capture, birds were randomly split into two housing groups (10 birds in each group, consisting of both males and females): (1) birds in standard laboratory cages (45 cm × 37 cm × 33 cm) outfitted with a water and food dish (hereafter referred to as ‘birds initially in standard cages’ or ‘Standard → Grass’ in figures) and (2) birds in cages with a water dish, but in lieu of a food dish, seed is spread across artificial grass to simulate a more natural foraging environment (hereafter referred to as ‘birds initially in grass cages’ or ‘Grass → Standard’ in figures). In both groups, birds were put on a light cycle of 12L:12D, and were provided access to ad libitum food (commercial mix of millet and sunflower seeds) and water throughout the experiment. Every cage was outfitted with multiple wooden perches and a pumice stone, as enrichment. Birds were doubly (M/F) or individually (M) housed, and were in auditory and visual contact with conspecifics throughout the experiment. After capture, birds were monitored for 3 weeks, which coincides with what we have observed in previous studies as a timepoint where changes in these physiological metrics have normally plateaued. To test whether foraging on artificial grass could serve as a temporary treatment to ameliorate the transition to captivity, all birds switched housing groups after this 3 week period (birds in standard cages were moved to artificial grass cages and vice-versa). Monitoring continued for 3 additional weeks. All experiments were approved by the Tufts University Institutional Animal Care and Use Committee. Portions of this text were previously published as part of a conference (https://s3.amazonaws.com/xcdshared/sicb/app_content/1526_1230033635.pdf).

Several metrics were measured throughout the experiment (Fig. 1). To ensure that all birds were successfully foraging, weight was measured twice weekly. To determine if there was a time-component to any physiological responses, blood was collected at the time of capture and continued once weekly throughout the experiment. Baseline blood sampling occurred within 3 min of the bird entering the mist net (day 0) or of the experimenter entering the laboratory housing room to ensure baseline hormone levels (Romero & Reed, 2005). If samples were collected after 3 min, they were excluded from baseline corticosterone analysis. Baseline samples were collected 1.5 h after lights turned on. ∼30 µL of whole blood was collected in heparinized capillary tubes following venipuncture of the alar vein. Two µL of whole blood were removed from the tube and stored at −80 °C until assayed for DNA damage quantification via the comet assay (Afanasieva & Sivolob, 2018). Samples from the time of capture were excluded from the comet assay due to UV exposure. Plasma was separated from all blood samples via centrifugation at ∼1,200 g and stored at −20 °C until assayed for corticosterone quantification.

Figure 1 Experimental timeline.

At the time of capture, birds were randomly assigned to two housing groups: cages lined with artificial grass (Grass) or standard laboratory cages with food dishes (Standard). Birds were monitored for 3 weeks before all individuals switched treatment groups and were monitored for 3 additional weeks. Symbols indicate date of sampling for the corresponding assay. Negative feedback efficacy (inverted V) was measured at weeks 3 and 6. DNA damage (blue diamond) was measured weekly from week 1. Baseline corticosterone (cort; red circle) was measured weekly from day 0. Weight (black square) was measured bi-weekly from day 0.

Additional blood samples were collected on day 21 of the experiment (the day birds switched housing groups) and day 42 (the final day of the experiment). We collected a sample for baseline corticosterone within 3 min of entering the room, then collected a stress-induced corticosterone sample after birds were restrained for 30-min in a cloth bag (capture stress protocol) (Wingfield, Smith & Farner, 1982). In addition, corticosterone negative feedback efficacy was determined by sampling 90 min following injection of synthetic glucocorticoid, dexamethasone (DEX) at a dose of one mg/one kg body weight (Cole et al., 2000) and calculated as the percent reduction from stress-induced corticosterone (Lattin & Kelly, 2020). At the end of the experiment, birds were used for other experiments in the lab (not part of the present study).

Comet assay

To quantify DNA damage, the comet assay was performed as described in Gormally et al. (2020). The protocol is performed under low light conditions to reduce exposure to UV-induced DNA damage. Samples collected in the field at day 0 were removed from analysis, as they were exposed to UV for extended periods of time. Briefly, two µL aliquots of whole-blood were diluted in 800 µL of phosphate-buffered saline (PBS). Then, two five-fold dilutions were performed (100 µL of suspension into 400 µL of PBS). 30 µL of the final sample dilution were added to 300 µL of low-melting agarose (Cat. # 425005001; R&D Systems), which was previously melted in boiling water and cooled to 37 °C . Samples were briefly vortexed and then 30 µL was plated on a 20-well CometSlide (Cat. #425005001; R&D Systems) in duplicate. After plating samples, slides were placed in 4 °C for 30 min to allow agarose to solidify. Then, slides were immersed in a pre-chilled lysis solution (Cat. # 425005001; R&D Systems), mixed with 10% dimethyl sulfoxide for 1 h at 4 °C followed by submersion in electrophoresis buffer (300 mM sodium acetate, 10 mM Tris base, pH 10) for 30 min at 4 °C. Slides were then electrophoresed at 21V for 30 min at 4 °C , then washed twice in dH2O and once in 70% ethanol for 5 min each. Slides were placed in a drying oven (37 °C ) until no agarose was visible (∼30 min), then stored in slide boxes with desiccant until staining and imaging. Slides were stained for DNA with SYBR Gold (Invitrogen, Cat. #S11494; 1:30,000 dilution in TE Buffer) for 30 min, then rinsed once with dH2O. Slides were again placed in the drying oven until no liquid was visible (∼15 min).

Slides were imaged using a confocal microscope (Leica Microsystems, LAS X Thunder) with a 488 nm laser at an objective of 10x. The OpenComet plugin (Gyori et al., 2014) for FIJI (Schindelin et al., 2012) was used to quantify the images. Aberrant comets were manually removed from analysis, and investigators were blind to housing group while running the OpenComet plugin. We used the TailMoment metric of DNA damage as in Gormally et al. (2020); Beattie et al. (2022). Intra- and inter-assay variation were 14.2% and 18.7%, respectively.

Corticosterone analysis

Plasma corticosterone was quantified using a competitive ELISA kit (Arbor Assays, Cat. # K014-H5), previously validated for use in house sparrows (Martin et al., 2017). For each sample, 5 µL of plasma was added to 5 µL of dissociation reagent and allowed to incubate for 5 min. Then, 240 µL of assay buffer was added to each sample and vortexed briefly. Samples and standards (39–10,000 pg/mL) were pipetted onto the plate in duplicate (50 µL per well) then incubated with corticosterone conjugate and antibody at room temperature (RT) for 1 h while being shaken at 700 rpm using a commercial microplate orbital shaker (FisherScientific, Cat. # 88-861-023). Then, plates were washed 4x with wash buffer provided by the kit, incubated with tetramethylbenzidine TMB substrate for 30 min, and stop solution was added to wells. Plates were immediately read using a microplate reader (Molecular Devices, SpectraMax M3) at 450 nm. Intra- and inter-assay variation were 9.6% and 13.9%, respectively.

Statistics

All statistical analyses were conducted in R version 4.2.2. For all variables with repeated sampling (weight, baseline corticosterone, negative feedback efficacy, and DNA damage), the effect of the variable was analyzed using a linear mixed effect model (‘lmer’ function, lme4 package) with group (birds initially in grass cages v. birds initially in standard cages), day, sex, and social housing status (individually or doubly-housed birds) as fixed effects and bird identity nested within cage ID as a random effect. Raw data values of the variables were used for all analyses. Then, all models were analyzed with a Type III analysis of variance (ANOVA) to test for significant effects. If there were significant effects, the model was split into two models by group: one for birds initially in grass cages and one for birds initially in standard cages. In this case, another linear mixed effect model was run to test variable with day as a fixed effect and bird identity as a random effect. Then, a Type III ANOVA was run to test the significance of these models. If there was a significant effect of day, data were further analyzed using multiple comparisons of means (‘glht’ function, multcomp package). This function was used to test for pairwise differences, comparing each day of the experiment to day 0 (day of capture) and day 21 (day of switching groups). For all variables, data were subset by day to make comparisons between housing groups on specific experimental days. Then, Wilcoxon rank-sum tests were performed. Prior to analysis, data sets were checked for homogeneity of variances through Levene’s Test and for normality using the Shapiro–Wilk normality test.

For negative feedback efficacy, the percent reduction from stress-induced cort were also analyzed using a two-way ANOVA with group and day as the effects, followed by a Tukey’s post-hoc test. To make paired comparisons, data was subset by group and a Wilcoxon signed-rank test was performed comparing day 21 and day 42 for each group.

Results

Weight

Both day (F11,220 = 44.8, p < 0.001) and the interaction between group and day (F11,220 = 8.8, p < 0.001) significantly affected weight in the linear mixed effects model (Fig. 2). Sex and social housing status did not have a significant effect (p = 0.27 and p = 0.34, respectively). For birds initially in grass cages there was a significant effect of day on weight (F11,110 = 29.1, p < 0.001). Post-hoc pairwise comparisons show that after introduction to captivity there was a decrease in weight as of day 10 (z =  − 6.26, p = <0.01) which persisted throughout the experiment (p = <0.01 for all other comparisons). Of note, after these birds were switched from grass cages to standard cages there was further decrease in weight (z = −4.43, p = <0.01) which persisted until day 35 of the experiment (z = −3.23, p = 0.056). For birds initially in standard cages there was a significant effect of day on weight (F11,110 = 23.1, p < 0.001). After introduction to captivity there was an immediate decrease in weight indicated by day 7 measurements, (z = −4.61, p = <0.01) which persisted throughout the experiment (p = <0.01 for all other comparisons). There was initially no further decrease in weight (z = 0.05, p = 1.00) after these birds were switched from standard cages to grass cages. However, there were two non-consecutive days where weight was significantly lower when compared to the last day before the switch: day 35 (z = −7.13, p = <0.01) and day 42 (z = −7.18, p = <0.01). When comparing housing groups using Wilcoxon rank-sum tests, weight was significantly lower in birds initially in grass cages compared to birds in standard cages only on days 24 (p = 0.007), 28 (p = 0.002), and 31 (p = 0.02).

Figure 2 Changes in weight following introduction to captivity.

Weight is plotted as percent of initial weight at time of capture. Weight decreased in both groups following introduction to captivity. Vertical dashed line indicates the day birds switched treatment groups. Stars (*) indicate significant differences from day 0 and pound symbols (#) indicate significant differences from day 21 (post-hoc multiple comparisons of means, p < 0.05). Symbols are above the line for Standard-Grass and below the line for Grass-Standard. Plus (+) symbols indicate significant differences between groups (Wilcoxon rank-sum test, p < 0.05). Bars indicate mean ±  standard error.

DNA damage

Day significantly affected the average tail moment of samples (F5,93 = 2.61, p = 0.03) in the linear mixed effects model (Fig. 3). Sex and social housing status did not have a significant effect (p= 0.14 and p = 0.31, respectively). When subdivided by group, day remained a significant factor for birds initially in grass cages (F5,46 = 4.67, p = 0.002), but not for birds initially in standard cages (F5,46 = 0.44, p = 0.82). Post-hoc pairwise comparisons show that for birds initially housed in grass cages, DNA damage increased on day 28 of the experiment (z = 3.45, p = 0.007) and remained elevated throughout the remainder of the experiment (day 35: z = 2.79, p = 0.001; day 42: z = 2.65, p = 0.048) when compared to measurements taken on day 7 of captivity. When comparing housing groups using Wilcoxon rank-sum tests, DNA damage was significantly lower in birds initially in grass cages compared to birds in standard cages on day 7 only (p = 0.04).

Figure 3 Changes in DNA damage following introduction to captivity.

DNA damage is plotted as the Tail Moment calculated from the comet assay, a measure of double-stranded breaks in DNA. DNA damage significantly increased for birds that switched from grass to standard cages (day: F5,46 = 4.67, p = 0.002). Dashed line indicates the day birds switched treatment groups. Stars (*) indicate significant differences from day 0 (post-hoc multiple comparisons of means, p < 0.05). Plus (+) symbols indicate significant differences between groups (Wilcoxon rank-sum test, p < 0.05). Bars indicate mean ±  standard error.

Baseline corticosterone

Day significantly affected baseline corticosterone (F6,96 = 2.62, p = 0.021) in the linear mixed effects model (Fig. 4), and there was a significant effect of group (F1,23 = 6.74, p = 0.016) as well as social housing status (F1,22 = 7.47, p = 0.012). There was no effect of sex (F1,21 = 2.70, p = 0.11). When subdivided by group, day was not a significant factor for birds initially in grass cages (F6,47 = 1.66, p = 0.15), or for birds initially in standard cages (F6,46 = 1.63, p = 0.16). When comparing housing groups using Wilcoxon rank-sum tests, baseline corticosterone was only significantly lower in birds initially in grass cages compared to birds in standard cages on day 21 (p = 0.04).

Figure 4 Changes in baseline corticosterone following introduction to captivity.

Baseline corticosterone (cort) measured from plasma samples throughout the experiment. Cort did significantly increase following introduction to captivity (day: F6,96 = 2.62, p = 0.021) and did differ between groups (group: F1,23 = 6.74, p = 0.016). Dashed line indicates the day birds switched treatments. Plus (+) symbols indicate significant differences between groups (Wilcoxon rank-sum test, p < 0.05). Bars indicate mean ± standard error.

Negative feedback efficacy

In the linear mixed effects model, day is significant (F1,35 = 6.62, p = 0.015), while there was no significance of group (F1,35 = 0.13, p = 0.71), social housing status (F1,35 = 2.62, p = 0.11), or sex (F1,35 = 0.57, p = 0.46). Negative feedback efficacy (Fig. 5) was higher on day 21 than day 42 (F1,11 = 5.74, p = 0.02, two-way ANOVA). Tukey’s post-hoc tests revealed no significance differences between group and days (p > 0.05). However, paired Wilcoxon signed-rank tests comparing day 21 and day 42 show that birds initially in cages with artificial grass increased negative feedback efficacy after moving into standard cages (p = 0.04). Whereas birds initially in standard cages had no change in negative feedback efficacy after being housed in grass cages (p = 0.23).

Figure 5 Negative feedback efficacy increased in birds initially acclimated to artifiical grass after switching to standard cages.

Negative feedback efficacy was determined as the relative reduction of negative feedback concentrations from stress-induced corticosterone concentrations (higher numbers reflect stronger negative feedback). Individual data points are shown as circle or triangles. Stars (*) indicate significance (Wilcoxon signed-rank test, p < 0.05). n.s., not significant. Bars indicate mean ± standard error.

Discussion

In the present study, we introduce the use of artificial grass as an alternative laboratory feeding procedure, as a potential environmental enrichment to reduce physiological stress in house sparrows during introduction to captivity. We monitored three established markers of a physiological response to captivity throughout the experiment: (1) body condition (weight), (2) DNA damage, and (3) circulating corticosterone concentrations. If successful as an enrichment, we predicted that birds in the artificial grass treatment group when compared to birds in standard cages would exhibit (1) weight loss to a lesser degree or at a slower rate, (2) lower amounts of DNA damage, and (3) no or less pronounced increases in baseline corticosterone concentrations, and (4) greater negative feedback efficacy. Our findings partially support these predictions.

Consistent with prior studies on house sparrows, all birds experienced significant weight loss (Fig. 2) in response to introduction to captivity. Weight did eventually stabilize by approximately Day 12, but at a level lower than that of initial capture, similar to previous studies (Love, Lovern & DuRant, 2017; Fischer, Wright-Lichter & Romero, 2018; Gormally et al., 2019b). Interestingly, after birds had acclimated to grass cages for three weeks and were switched to standard cages, weight loss continued, whereas weight remained initially stable for birds that were switched from standard to grass-lined cages. This meant that weights remained significantly lower in birds that were initially given grass. These data suggest that if birds acclimate to a laboratory while provided the artificial grass treatment, switching them to a standard cage could cause further detrimental effects on body condition. Thus, the artificial grass treatment may buffer weight loss and is not suitable for only a transitional period and would have to be provided longer-term.

To assess DNA damage in the blood, the comet assay was performed using baseline samples. This assay measures double-stranded breaks in DNA and damage increases in response to both acute stress (Beattie et al., 2024; Malandrakis et al., 2016), and chronic stress, specifically as early as 3 days after introduction to captivity (Gormally et al., 2019b). It is important to note that in the present study we excluded day 0 measures of DNA damage due to likely UV exposure during field sampling, which is known to cause damage to erythrocytes (Afanasieva & Sivolob, 2018). That means we may have missed a key time window of increasing DNA damage as a response to captivity introduction. Despite this, significant differences in DNA damage between housing groups were observed (Fig. 3). Damage was significantly lower in birds that were housed in cages with artificial grass compared to birds housed in standard cages after weeks 1 and 2 of the experiment. This suggests that providing food on artificial grass may have buffered these individuals from accumulating DNA damage during introduction to captivity. Furthermore, when birds housed in grass cages were moved to standard cages, we observed significant increases in DNA damage. Although it is unclear whether the post-switch increase in damage was caused by the switch or simply extended a rise already in progress by Day 21, the data nevertheless provide further evidence that once acclimated to feeding from artificial turf, there may be detrimental effects associated with then switching to standard laboratory cages with food dishes.

Overall, there was a significant effect of group on baseline corticosterone (Fig. 4), indicating that birds initially in the grass treatment exhibited lower circulating corticosterone. However, when data were compared by day in a pairwise manner, day 21 was the only day wherein baseline corticosterone was significantly lower in birds housed in grass cages compared to standard cages; on all other days of the experiment, there were no significant differences between housing groups or changes in concentrations. Therefore, the effect of group was likely driven by this one day. Nevertheless, this provides some evidence that after 3 weeks, the treatment may have aided in reducing circulating corticosterone. Moreover, singly-housed birds had lower baseline corticosterone when compared to doubly-housed birds, which is surprising given that doubly-housing is a welfare standard.

Surprisingly, while baseline corticosterone did significantly increase for all birds throughout the experiment (Fig. 4), this significant effect was lost when data were subset by housing group. These data are not consistent with several studies wherein baseline corticosterone increased in house sparrows and other avian species during introduction to captivity (Fischer, Wright-Lichter & Romero, 2018; Fokidis et al., 2011; Love, Lovern & DuRant, 2017). While this finding was surprising, it is not completely unexpected as an analysis of multiple studies of chronic stress revealed that while baseline corticosterone usually increases in response to captivity, there are several cases where this is not a consistent pattern of change in baseline corticosterone concentrations (Dickens & Romero, 2013). Moreover, a recent meta-analysis investigating the measure of glucocorticoids in relation to welfare outcomes revealed inconclusive results (Tiemann et al., 2023), further emphasizing the need for researchers to include additional physiological measures during studies, and not simply rely on baseline corticosterone concentrations.

On day 21 (prior to switching housing groups) and day 42 (the final day) of the experiment, additional corticosterone measurements were collected to assess negative feedback. Negative feedback efficacy provides valuable insight into an individual’s ability to effectively regulate a stress response (Cole et al., 2000; Lattin & Kelly, 2020). Prior studies indicate that stronger negative feedback efficacy predicts survival (Romero & Wikelski, 2010), predicts resilience to future challenges (Zimmer et al., 2019), and correlates with reproductive success (Vitousek et al., 2019). Therefore, negative feedback efficacy is a good predictor of the future success of an individual or group, and in turn a good measure of wellness. Negative feedback efficacy did increase over the course of the experiment (Fig. 5), suggesting that birds were adjusting to captive conditions. Moreover, there was a significant increase in negative feedback efficacy from day 21 to day 42 in birds that were initially housed in grass cages and were switched to standard cages, which may indicate that moving to standard cages increased their ability to regulate a stress response. In contrast, our data show that birds that moved from standard to grass housing did not show an increase in negative feedback efficacy (Fig. 5), suggesting that artificial turf provided no major fitness advantage.

When taken together, these data provide limited evidence that our modification to the captive environment alleviated the physiological changes associated with introduction to captivity, especially given the potential for Type II error associated with repeated measuring, and an uneven mix of sexes. Birds provided food on grass-lined cages during introduced to captivity were seemingly buffered from accruing DNA damage (Fig. 3), but similar buffering was not seen for other metrics such as weight loss (Fig. 2), baseline corticosterone (Fig. 4) or negative feedback efficacy (Fig. 5). Moreover, when birds were switched from grass-lined cages to standard cages with food dishes, we observed further weight loss (Fig. 2) and increases in DNA damage (Fig. 3)—similar to a response characteristic of an initial introduction to captivity. This finding provides evidence that if birds acclimate to feeding on artificial grass, there could be stress-induced detriments to the physiological state of a bird if subsequently transitioned to standard cages, even temporarily. Additionally, these birds that had acclimated to captivity in grass-lined cages increased negative feedback efficacy after being housed in standard laboratory cages, indicating that they were better able to regulate a stress response in the standard cages.

Considering the logistics of animal husbandry, it is important to note that the artificial grass treatment was more difficult to maintain compared to standard cages. When soiled, commercially available cage liners are easy to dispose of and replace with clean liners. In contrast, the artificial grass needed to be removed, manually washed, and allowed time to completely dry when soiled. Logistically, researchers would need at least 2x the amount of artificial grass than the number of active cages for continuity of the treatment. Additionally, artificial grass is not likely to survive the high temperatures of commercial cage washers following recommendations by both the Guidelines to the use of wild birds in research (Fair, Paul & Jones, 2010) and the Guide for the Care and Use of Laboratory Animals (National Research Council, 2011), adding to the amount of additional time necessary to maintain this housing environment. For future research, we encourage researchers to test enrichments that are easy to maintain and sanitize, yet remain relevant to the species’ behavioral biology, such as providing sparrows with shelter, allowing them to forage under cover or providing a variety of branch types for the birds to perch upon. Given the limited evidence of beneficial effects, observed consequences of removing the grass-lined cages, and logistical complications of this cage setup, we do not recommend that other researchers use this feeding method as an enrichment without further experimentation such as incorporating a behavioral metric into the assessment or designing a preference system to test the enrichment. For future studies, we encourage researchers to test other novel forms of enrichment, particularly ecologically relevant ones, and incorporate physiological measurements as part of the assessment.

Conclusions

To address a need for experiments that test the usefulness of potential forms of enrichment that reduce stress in captive laboratory animals, we tested a system of providing feed to birds by spreading it across artificial grass instead of in standard food dishes. We monitored established biological markers known to change in response to chronic captivity stress—weight, DNA damage, baseline corticosterone concentrations, and negative feedback efficacy. We hypothesized that altering the foraging strategies of captive birds by providing food on artificial grass would serve as a form of enrichment and mitigate these physiological consequences of introduction to captivity. However, although the grass-housed birds had lower DNA damage, there were no notable differences in any of the other physiological measures. Further, when grass-housed birds were switched to standard cages, weight decreased, and DNA damage increased. Furthermore, negative feedback efficacy actually increased after birds were switched to standard cages. We conclude that this study provides insufficient evidence to recommend artificial grass as a suitable enrichment for captive birds, especially given the logistical challenges of implementing this treatment.

Supplemental Information

Supplemental Information 1 Raw data and code

All raw data used to generate figures and perform statistical analyses and an R script used to analyze data.

The raw data shows measurements of weight, corticosterone concentrations and DNA damage quantification for all birds throughout the experiment. These data were used for all statistical analyses.

Supplemental Information 2 ARRIVE Checklist

We thank Ursula Beattie, Rachel Riccio, Emma Rosen, and Paul Jerem for assistance in sample collection. We also thank the Tufts University animal care staff.

Additional Information and Declarations

Competing Interests

Author Contributions

Animal Ethics

Data Availability

The authors declare there are no competing interests.

Bradley P. Pedro conceived and designed the experiments, performed the experiments, analyzed the data, prepared figures and/or tables, authored or reviewed drafts of the article, and approved the final draft.

L. Michael Romero conceived and designed the experiments, performed the experiments, authored or reviewed drafts of the article, and approved the final draft.

The following information was supplied relating to ethical approvals (i.e., approving body and any reference numbers):

All experiments were approved by the Tufts University Institutional Animal Care and Use Committee.

The following information was supplied regarding data availability:

The raw data and R code used to generate statistical analyses are available in the Supplementary File.

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
