# Peer review of "Physiological responses of wild birds to artificial grass during introduction to laboratory housing"

_PeerJ, doi:10.7717/peerj.19095_

## Round 0.1 · original submission · Major Revisions

· Academic Editor

Major Revisions

I apologize for the extended reviewing time period. Sometimes it takes a while to collect sufficient reviews. Please pay close attention to each point made by the reviewers and, when resubmitting your article, send a point-by-point response - making it clear where in the manuscript you have made changes.

Reviewer 1 ·

Basic reporting

The manuscript is well written with the sufficient background information and proper literatures cited

Experimental design

The research question is well defined and methods are clearly described with the sufficient information to replicate

Validity of the findings

The data is statistically sound and controlled

Additional comments

The manuscript from Pedro and Romeo examined the effect of artificial grass as an enrichment method to reduce stress due to the introduction to captivity in wild-caught house sparrows. The authors found that both groups initially lost weight, but the grass-caged birds continue to lose more weight after switching to standard cages. In addition, DNA damage was lower initially in grass-cage birds but again increased once switching to standard cages. Interestingly, corticosterone levels were not different between groups. The authors concluded that artificial grass is not a strong enough enrichment to reduce the stress associated with captivity. This work provides valuable insights on the effect of enrichment strategy on stress response. However, some interpretation of the data needs to be addressed.

Major comments
1. The main point of this study is to test effect of artificial grass on the physiological parameters in the wild-caught sparrows. However, there are 2 parameters introduced in the enrichment groups, artificial grass and feeding method. The authors did not test effect of adding just artificial grass or scattering food on the cage floor independently. Therefore, any effects observed in this study could be the result from a single factor or combined factor of grass and feeding method. It would be more accurate to adjust the title to include or emphasize the foraging method factor as well.

2. Is there any sex difference in the response to the enrichment? Do both groups contain some of the female sparrows? This information should be included in the materials and methods section.

3. The authors mentioned (line 210-211) that the corticosterone values from day 0 was removed from the model because the birds had not yet undergone treatment yet. The day 0 value serves as an important baseline for all the birds and should be used to compare with the later time point to determine the effect of treatment.

4. In Figure 4, the corticosterone levels in both groups are not different in day 0 and keep increasing on day 7 and 14. However, on day 21, there is a significant drop in corticosterone in the grass group. This could indicate that grass help reduce corticosterone level. Have the authors considered this possibility?

5. In Figure 5, there is a high variation within the group and therefore could mask any effects of the treatment. It would be informative to do a paired comparison of the negative feedback efficiency in the same bird between day 21 and day 42 to determine if the efficiency increase or decrease after switching the group.

Minor comments
1. The authors should consider adding symbols to indicate any significant different between treatment groups in figures.
2. Line 157: cort should be corticosterone, there is no establishment of cort abbreviation until the legend in the Figure 4

Reviewer 2 ·

Basic reporting

The manuscript meets all the criteria for basic reporting. It is well-written and organized, and rooted in the appropriate literature regarding chronic stress and physiological markers in birds. The figures are clear and raw data accessible.

Suggestions to improve Figure 2: I wonder if it is useful to highlight all timepoints where the weight differed significantly from initial capture (as all points on the graph do), and instead highlight periods where the two treatment groups differed, since there was a significant interaction? The specific days where the treatments differed are mentioned in text on lines 192-194, are these worth adding to the figure?

Note also that statistics for the overall effect of treatment on baseline CORT are included in the Figure 3 legend but not in text, I recommend placing them in text as well.

Experimental design

The research question fills a gap in our knowledge about how interventions to improve captive animal welfare, particularly foraging enrichment, can affect physiological correlates of stress in wild birds. Such physiological tests are needed, as most assessments rely on behavioral metrics. The goal, hypothesis, and predictions of the study are well described.

While the Methods are clear overall, they could benefit from clarification of the rationale for switching captive environments. Based upon the hypotheses, the standard cages serve as a control for the artificial grass treatment. Why the need to switch environments? In the Discussion, one infers this is because researchers might be using artificial grass only temporarily to ameliorate the transition to captivity. If the authors could describe their rationale or goals for this design in the Methods section, that would be helpful.

Validity of the findings

The authors clearly linked the results to their hypotheses and predictions. I appreciated that the conclusions included practical recommendations for use of this enrichment in a captive setting. The authors also address some of the weaknesses of their design (such as lack of baseline DNA damage measures) and surprising results (i.e, no difference in baseline CORT).

Data have been provided, though in order to run the code for the Figures for weight (Figure 2), a file "HOSP Foraging Weights.csv" is currently missing from the supplemental files.

·

Basic reporting

This is a welcome first look at a very deserving, important topic that we hope will inspire future research on how best to house wild animals in research labs.

However, it does need some work, especially for clearer logic and more transparent conclusions: the background rationale (in terms of ethics, research model validity and welfare science) is very sparse; the hypotheses and their predictions really aren’t clear until the Discussion; the ‘cross-over’ design is never justified or explained (and thus remains something of a puzzling mystery even to the end!), and some other elements of the experimental design are hazy too; and the conclusions do not seem to follow from the results.

More details are provided below.

Rationale
i) Why care about wild animal welfare? The very background as to the important of research projects like this should be mentioned, even if only briefly. For example, are there regulatory reasons to do so? Is it a matter of living up public trust in science? Or does working with stressed animals compromise research / data validity?
ii) Why might wild animals be stressed in captivity, and how might ‘enrichments’ (a terrible term) address that? Some basic biology is missing here, in terms of animals’ evolved habitat preferences, and their evolved innate motivations to behave in certain ways. Furthermore, better scientific reasons are needed for the choice of ‘enrichment'. Many of the references provided are for lab rodents, and some of the outcome measures presented are also irrelevant for welfare (who care if rodents are more active, for example?). Perhaps closer is some of the evidence provided for parrots, but abnormal stereotypic behaviour is strangely discounted as a welfare indicator (when it is well known to be one); and furthermore, parrots are in completely different order (diverging from passerines for 10s of millions of years), and have very different foraging niches. The key issues here should be: how do sparrows live and forage in the wild? (Do they really forage alone, on turf, without any cover, for example?) What evolved habitat/behavioural needs are they likely to have (especially if separated from their normal flocks)? And what studies have there been of welfare in other captive finches (e.g. zebrafinches or canaries, if sparrow work is missing)?
iii) Line 61: Why are physiological data on welfare more important than behavioural? (To us, this paper’s lack of behavioural data, and poorly justified physiological measures, were actually a major weakness)

Hypotheses and predictions
What you would expect to see if grass does indeed improve welfare (and why these measures were chosen) is unclear until the Discussion (lines 226-230): really the wrong place to put predictions. Also they are not fully justified with any rationales/citations. Furthermore, interpretations of baseline cort are hard, which is why the authors quite rightly seem very equivocal (lines 265-267). In our own lab we simply never use baseline cort to assess welfare, because there is so much evidence (including from the authors’ own lab.!) that when subjects are placed longterm in aversive conditions, baseline cort. can increase, decrease or show no change at all (and NB. Isn’t Abstract line 16 incorrect? Cf the authors’ own starling work?). This means that almost any welfare story can be constructed from any detected pattern: obviously not what one wants to see in an informative indicator. Why not just chuck out this measure, since it has so little merit?
Also unclear is a lack of predictions around the crossover design: presumably if grass benefits welfare, then birds given grass will show less of a welfare hit when first taken captive, and birds who have grass taken away from them will show more signs of welfare harm that birds who, 3 weeks in, are then given grass? Spelling this out earlier, and structuring the Results, accordingly would make the paper much easier to read.

Experimental design

Experimental design and subject details
Why were birds from this social species kept on their own? Implications for the ethics and validity of this study?
How old were the birds?
What type of environment were they captured from? (e.g. urban versus rural: this could affect how aversive they find humans)
Why were birds given a housing switch at 3 weeks?
Since you only needed to assess the impact of 3 weeks of differential housing, why weigh them biweekly? Seems inefficient and needlessly stressful to the birds, since you don’t have predictions about her trajectory of change: only that there’d be differential weight loss after N weeks of differential housing. (Also, how was weighing conducted?). Surely you only needed data on Days 1, 21 and 42?
Likewise, why blood sample every week? Same concerns as above (but also, it means that many comparisons were run which increases the risk of Type I errors).
Why no baseline values for DNA damage: this was quite the oversight, no?

Analyses
Why wasn’t ‘sex’ included in the statistical model?
Weight was calculated as a % of starting weight (e.g. see Fig 2). Everything else was presented in terms of absolute values over time. Why the difference? Logic very unclear. Shouldn’t all data be treated the same for clarity?
Does ‘treatment’ (line 158) mean ‘current treatment’, or ‘grass first then none’ versus ‘no grass, then grass’? And was treatment*day in the model? It’s incredibly hard to follow what you did (both in the Methods and the Results), even though the research question (is grass good for sparrows?) is quite simple.
Does the ‘multicomp’ package correct for multiple testing? (ie adjust P values accordingly?)
How did you check for normality and homogeneity of residuals? Did data ever need transforming?
Line 210: Why remove Day 0 from the model? - seems very ad hoc (all part of the lack of clarity of hypothesis-testing).

Validity of the findings

See above queries re baseline cort.

Validity of conclusions:
Although it's hard to be sure, it seems you have evidence that removing grass causes weight loss, that providing grass reduces DNA damage, so surely you should be recommending grass as an 'enrichment'?
If you're not sure, how about suggesting some more sensitive tests to re-ask the question, such as preference studies?
And if you have practical qualms (though honestly, that you need multiple pieces of turf to allow for cage-cleaning -- as pointed out at lines 298-299 -- is a 'no-brainer' rather than a 'deal-breaker'!), why not suggest for future investigation other types of 'enrichment' that researchers might have better luck with (perhaps better rooted in sparrows' evolved behavioural biology), such as allowing them to forage under cover, and always housing with a flock-mate?

Additional comments

I refereed this with two graduate students in my lab (with permission from the journal).

---

## Round 0.2 · Major Revisions

· Academic Editor

Major Revisions

The decision of "Major Revision" is because some work still needs to be done on the stats and the interpretation of the data, along with a slight change in the title. Although it is unfortunate that neither I nor reviewer 3 picked up on the points that are now being made during the first round of review, I think that they are valid but addressable. Please pay careful attention to all the points made by Reviewer 3, but especially to the statistical analysis including cage as a factor. In addition, the fact that the birds that were initially provided with artificial turf but then had it removed were the only ones that continued to lose mass and yet showed increased negative efficiency feedback seems to imply (although can't be shown by the current data) that the initial provision of turf had some buffering effect in terms of the stress of initial transfer to captivity. Then the turf removal seems to have some other effect, although the data are quite variable. I think you have discussed this as much as you can, based on the data (in the absence of a 3rd experimental group which was supplied with turf throughout and a 4th which never had turf), but I agree with Reviewer 3 that the title should be adjusted to be less dogmatic (or less specific). Perhaps "Physiological responses of wild birds to artificial grass in laboratory housing" or suchlike (I'm not expecting you to use this example, it's just an example of one that might fit alternative interpretations of the data.

Overall, I think the paper is already in better shape after the first round of revision and your responses to Reviewer 3's comments will improve it further. I look forward to receiving your revised manuscript.

Reviewer 1 ·

Basic reporting

The authors have addressed and adjusted the title as suggested.

Experimental design

The authors clarified all the comments on the experimental design.

Validity of the findings

The authors have addressed the statistical analysis concern I had earlier.

Additional comments

The revised manuscript addresses all the concerns I had and I have no further comment or concern on the manuscript.

Reviewer 2 ·

Basic reporting

The manuscript meets all the criteria for basic reporting. It is well-written and organized, and rooted in the appropriate literature regarding chronic stress and physiological markers in birds. The figures are clear and raw data accessible.
In revision, the authors improved the data visualization to include significant effects of day but also interactive effects between the two housing environments.

I have no further suggestions

Experimental design

The goals, hypotheses, and methods of the experiment are well detailed. In revision, the authors added important experimental design details, such as sex in models and sex distribution across groups, in response to other reviewers’ comments. The authors also sufficiently justified their choice for a “cross-over” design, where birds switched from artificial grass to standard housing, or vice versa, and added hypotheses specific to that design.

I have no further suggestions.

Validity of the findings

The authors linked results to their initial hypotheses, and sufficiently addressed limitations of their design and/or methods. Importantly, in revision, the authors included data (HOSP Foraging Weights.csv) that was missing to allow the analyses to be reproducible.

I have no further suggestions.

Additional comments

The authors have adequately addressed all of my previous comments, as well as those from other reviewers.

·

Basic reporting

It's good to see the paper again, and the logic and experimental design are now much clearer which is great. However, there is still some opaque logic (sorry); and also, now that the experimental design is clearer, we're afraid the stats need a revision.

As one example of the former, references and justifications are still missing to support your use of your chosen variables as welfare indicators. For example, what is the prediction for the negative feedback measure, if turf is effective at improving welfare? The hypotheses and predictions are now in the Intro, which is great, but that one is missing. Also, can you also add references here, justifying your assumptions that weight loss to a lesser degree or at a slower rate, lower amounts of DNA damage, no or less pronounced increases in baseline corticosterone concentrations, and greater negative feedback efficiency, are all unambiguous signs of improved welfare? (Obviously you won't be able to do this for baseline cort! And if negative feedback efficiency is not actually a valid welfare indicator, it should actually be dropped).

Having upfront rationales and directional predictions like this, and ones that are backed by references, is important in welfare/policy-related work, in order to prevent HARKing.

Experimental design

Now that it's clear that some animals were pair-housed and some were single-housed, this mean that 'social housing status' needs to be a blocking factor in the model (a fixed effect just like sex, because, just like sex, being housed alone will influence stress-related outcomes). Furthermore, the two birds sharing a cage and the same 'enrichment' are non-independent (i.e. they will influence each other, and also literally share the same treatment making 'cage' the unit of replication; cf. e.g. https://journals.plos.org/plosbiology/article?id=10.1371/journal.pbio.2005282). 'Cage' therefore needs to added to all models as a random effect; without this, your stats are pseudoreplicative.

Also, are you investigating the benefits of 3 weeks of turf provision (for which you have data from all birds), or the costs of removing it after 3 weeks (for which you only have data from half the birds)? Right now, the confusing new statement that "we additionally tested if artificial grass could serve as a temporary enrichment for the purpose of ameliorating the transition into captivity by switching birds into standard cages after they acclimated to grass-lined cages" (at line 111) honestly feels like a post hoc explanation for a possibly almost accidental study design: who would really think that a sensible way to improve wild animal welfare?

(Relatedly, to investigate the benefits of 3 weeks of turf provision, could you perhaps pool Phase 1 of the 'turf first' group, and Phase 2 of the 'turf second' group [ with phase as a block])? Your current analyses fail to capitalise on the statistical power you could get by models set up to recognise that all birds experience this treatment.... though the lack of a proper control group in such an analysis is a problem we can't quite figure out how to fix. Otherwise, maybe it's a matter of presenting the data more clearly like that [since Phase 1 of the 'turf first' group, and Phase 2 of the 'turf second' group are so intrinsically similar]).

Validity of the findings

The findings are't valid in their current form, partly because of the pseudo-replicative stats and also because the construct validity of the measures chosen as welfare indicators is not justified (see above). We've said quite a bit in the earlier sections, but to add more in this vein:

Since birds initially given grass but then deprived of it show more weight loss and more DNA damage, that surely means it was a loss to them, i.e. that it did actually have value?

Also, this study was almost certainly underpowered, what with the mix of sexes, social housing statuses, and the strange cross-over design: as far as we can tell there were just 15 cages, with 7 in one group, 8 in another. So, a frank admission of the strong chances of Type II error is essential, especially as you're determined to use these data to try and make policy recommendations.

Correspondingly, the MS title also needs to be toned way down: it is currently overly strong considering all these issues.

Additional comments

A few last things:

1) Justifying measures or calculations by them being "very common in our field" is not really a justification. For one thing, some fields never use blinding or randomization, but obviously that doesn't make it right! Furthermore, in this MS you've moved from wild animal physiology to animal welfare assessment/work aimed at practical recommendations, which means meeting best practice expected there, in that (our) field (namely clear logic, measures with good construct validity for inferring welfare, and good statistical power).

2) Second, you really did not need all those multiple measures over time, since questions about time-courses really are not relevant for your research aims; and since they were stressful for the animals, we think you should flag that as an ethical weakness of the study (and tell your IACUC too: to monitor wild animal stress by catching them and daily stuffing them in a bag is truly insane! Install perches with strain gauges/load cells so you can simply see their weights when they sit!).

3) Surely providing companionship should be the number one recommendation for future work in your conclusions (a line c. 400)? Isolating social animals is almost always a chronic stress (see many many many studies).

Best wishes
Georgia Mason and two grad students

PS You said you couldn't find much else on finch/sparrow welfare. These papers are old, but pretty great:
Keiper, R. R. (1970). Studies of stereotypy function in the canary (Serinus canarius). Animal Behaviour, 18, 353-357.
Keiper, R. R. (1969). Causal factors of stereotypies in caged birds. Animal Behaviour, 17, 114-119.
Sargent, T. D., & Keiper, R. R. (1967). Stereotypies in caged canaries. Animal Behaviour, 15(1), 62-66.

Also if useful at all, I used in a cross-over design in this hen welfare study of effects of housing quality (using it to enhance power): https://www.nature.com/articles/s41598-019-48351-6; and this is another paper showing that losing 'enrichments' is worse than never having them: https://www.sciencedirect.com/science/article/abs/pii/S0166432810001889 (I actually don't love this paper but at least you might find someone of our cited papers worth chasing up).

---

## Round 0.3 · accepted · Accept

· Academic Editor

Accept

I have assessed your revised manuscript myself, and in my opinion you have addressed all of the reviewers' comments satisfactorily. This manuscript is now ready for publication.